# Challenges and the Evolving Landscape of Assessing Blood-Based PD-L1 Expression as a Biomarker for Anti-PD-(L)1 Immunotherapy

**DOI:** 10.3390/biomedicines10051181

**Published:** 2022-05-20

**Authors:** Tao Wang, Desirée Denman, Silvia M. Bacot, Gerald M. Feldman

**Affiliations:** Office of Biotechnology Products, Office of Pharmaceutical Quality, Center for Drug Evaluation and Research, Food and Drug Administration, Silver Spring, MD 20993, USA; desiree.denman@fda.hhs.gov (D.D.); silvia.bacot@fda.hhs.gov (S.M.B.); gerald.feldman@fda.hhs.gov (G.M.F.)

**Keywords:** immune checkpoint inhibitor, anti-PD-(L)1 immunotherapy, liquid biopsy, biomarker, PD-1, PD-L1, circulating tumor cells, circulating immune cells, exosomal PD-L1, plasma PD-L1

## Abstract

While promising, PD-L1 expression on tumor tissues as assessed by immunohistochemistry has been shown to be an imperfect biomarker that only applies to a limited number of cancers, whereas many patients with PD-L1-negative tumors still respond to anti-PD-(L)1 immunotherapy. Recent studies using patient blood samples to assess immunotherapeutic responsiveness suggests a promising approach to the identification of novel and/or improved biomarkers for anti-PD-(L)1 immunotherapy. In this review, we discuss the advances in our evolving understanding of the regulation and function of PD-L1 expression, which is the foundation for developing blood-based PD-L1 as a biomarker for anti-PD-(L)1 immunotherapy. We further discuss current knowledge and clinical study results for biomarker identification using PD-L1 expression on tumor and immune cells, exosomes, and soluble forms of PD-L1 in the peripheral blood. Finally, we discuss key challenges for the successful development of the potential use of blood-based PD-L1 as a biomarker for anti-PD-(L)1 immunotherapy.

## 1. Introduction

Anti-programmed death-1 (PD-1) or anti-programmed death ligand 1 (PD-L1) immunotherapy (anti-PD-(L)1 immunotherapy) has achieved unprecedented clinical efficacy for patients with various types and stages of cancers. The U.S. Food and Drug Administration (FDA) has approved four anti-PD-1 antibodies (nivolumab, pembrolizumab, cemiplimab, and dostarlimab) and three anti-PD-L1 antibodies (atezolizumab, durvalumab, and avelumab) for the treatment of nearly all types of cancers [1,2,3,4]. Despite these advances in cancer treatment, only a small subset of cancer patients actually benefits from anti-PD-(L)1 immunotherapy either due to tumor cells not responding to the therapy (primary resistance) or due to tumor cells developing resistance to the therapy (acquired resistance) [1,5,6,7,8,9]. Even worse, treatment with anti-PD-(L)1 immunotherapy can lead to an acceleration of tumor growth, defined as Hyperprogressive Disease [10,11,12]. Moreover, many patients terminate treatment because of therapy-induced severe toxicities that are associated with side effects caused by immune activation, referred to as immune related adverse events (irAEs) [13,14]. Therefore, identification of biomarkers to stratify patients who are most likely to benefit from the anti-PD-(L)1 immunotherapy is of critical importance. 

Of the three FDA-approved predictive biomarker tests for anti-PD-(L)1 immunotherapy, PD-L1 assessed in tumor tissues by immunohistochemistry (IHC) is the most commonly used biomarker for patient stratification [15,16]. The biological rationale for this is based on the mechanism of action of anti-PD-(L)1 immunotherapy, in which generation of the effective anti-tumor specific T cell immune response requires two signals: (**1**) T cell priming provided by the T cell receptor (TCR) recognizing and binding to antigen peptide in complex with human leukocyte antigen (HLA) molecules. Along this line, non-germline mutations in TMB-H (high tumor mutation burden) and/or MSI-H (high microsatellite instability) tumors generate tumor-specific neoantigens. This provides a primary signal by forming neo-antigen peptide/HLA complexes that are recognized by TCR, resulting in initiating activation of tumor-specific T cells [17,18,19]. (**2**) Full T cell activation provided by engagement of co-stimulatory receptors expressed on T cells and its ligands expressed on antigen presenting cells (APCs), such as co-stimulatory receptor CD28, and its ligands CD80 (B7-1) and CD86 (B7-2). PD-L1 expressed on tumor cells and/or tumor stromal cells impedes this second signal by engagement of PD-1 expressed on T cells, resulting in dysfunction of anti-cancer T cell responses [20,21,22,23]. 

PD-L1 expression on tumor tissues has clearly shown the predictive value in many types of cancers, as patient responses to anti-PD-(L)1 immunotherapy are linearly associated with increased levels of PD-L1 expression in many types of cancers [24,25,26]. However, positive PD-L1 expression can only partially predict which patients benefit from therapy, as a subset of patients whose tumors lack expression of PD-L1 has also been shown to respond positively to anti-PD-(L)1 immunotherapy. Moreover, most cancer patients do not achieve a durable response regardless of levels of PD-L1 expression [15,27]. The inability of tumor biopsy to capture the complexity of intra-tumor heterogeneity and heterogenous phenotypes in tumor tissues (spatial limitation) is a major attribute of the imperfection of PD-L1 use as a biomarker [28,29,30]. Additional challenges to this approach include difficulties in acquiring tumor biopsies, such as early stage pancreatic cancer tissues and their unsuitability for longitudinal monitoring of the dynamic changes of tumor cells and other stromal cells within the tumor microenvironment (temporal limitation) [27,31]. Lastly, it is challenging to harmonize the assays to assess PD-L1 expression in tumor tissues, including how to select an appropriate PD-L1 assay platform and how to score the PD-L1 expression consistently and accurately on tumor cells, immune cells, or both [32,33,34]. An alternative approach, using liquid biopsies to analyze PD-L1 expression on cytological samples has shown promise, which has the potential to overcome many of the challenges observed when using solid tumor biopsies [35,36]. 

While enormous efforts have been made for improving the quality and accuracy of PD-L1 assessments using tumor biopsy-based approaches [37], recent advances in liquid biopsy might not only overcome the many hurdles facing the field of identification of biomarkers in tumor biopsies, but also provide opportunities to discover novel biomarkers or to improve the accuracy of currently approved biomarkers. The rationale for using blood-based biopsies for identifying biomarkers is that nearly all components in the tumor microenvironment can be found in blood. These components include genomic material, proteins, metabolites, and extracellular vesicles, as well as various cell types including circulating tumor cells (CTCs) and immune cells trafficking between tumors to the blood stream and lymphoid organs [38]. In comparison with tumor biopsy, a blood-based biopsy approach has the following advantages: (**1**) the convenience of sampling can be used to longitudinally monitor the dynamic biological changes within the tumor microenvironment; (**2**) samples in the blood contain components derived from both primary and metastatic tumor tissues; (**3**) ease of development and validation of many assays for testing multiple blood components; and (**4**) systemic anti-tumor T cell responses have recently emerged as a promising potential biomarker, which can only be tested using peripheral blood mononuclear cells [39]. 

In this review, we provide an overview of recent advances in understanding the complexity of regulation and function of PD-L1 expression. We discuss the rationale for using different components of blood to assess PD-L1 as a biomarker with a focus on the mechanistic and technical challenges involved. Lastly, we discuss the key challenges and prospects in the development and validation of blood-based PD-L1 assessments for anti-PD-(L)1 immunotherapy based on our evolving understanding of the mechanisms of action underlying this therapy. 

## 2. Regulation and Function of PD-L1 Expression

PD-L1 (B7-H1, CD274) is a type 1 transmembrane glycoprotein encoded by the CD274 gene. PD-L1 is highly expressed on tumor cells, as well as on hematopoietic cells, including macrophages, dendritic cells, mast cells, neutrophils, myeloid-derived suppressive cells (MDSCs), platelets, and T and B lymphocytes, and on non-hematopoietic cells, such as epithelial cells, endothelial cells, mesenchymal stem cells, placental trophoblasts, and many others [15]. Expression of PD-L1 is regulated at the transcriptional, post-transcriptional, and post-translational levels [40,41,42,43]. At the transcriptional level, PD-L1 expression is modulated by JAK/STAT1, NF-kappa B, and other transcriptional factors. Additionally, glycosylation and acetylation of PD-L1 are two important post-translational regulators of PD-L1 expression. For example, acetylation of PD-L1 impacts the translocation of PD-L1 from the cytoplasm to the nucleus, whereas glycosylation of PD-L1 can enhance its ability to interact with PD-1 to increase anti-cancer T cell immune responses [44,45]. Expression of PD-L1 is also impacted by protein degradation pathways such as autophagic degradation and ubiquitination. PD-L1 degradation pathways not only affect the efficacy of anti-PD-(L)1 immunotherapy, but also affect its pharmacokinetics and pharmacodynamics—parameters that are critical for the timing of sampling of PD-L1 for biomarker assessment [46,47,48]. The regulation of PD-L1 expression differs significantly between different cancer types, as well as between tumor cells and non-tumor cells due to variation of aberrations of activation of many signaling pathways such as RAS/RAF/ERK, PI3K/AKT, and JAK/STAT3, among others [41]. Of note, gene amplification, gene translocation, copy-number gains, or copy-number loss also occur in tumor cells [47,49]. PD-L1 copy number changes are associated with PD-L1 expression in several types of cancers such as NSCLCs, urothelial carcinoma, and breast carcinoma, among others. However, PD-L1 gene amplification is not associated with PD-L1 expression when tumor tissues are assessed by IHC in a subset of Hodgkin lymphoma patients [50,51,52,53].

The major function of PD-L1 is suppression of T cell activation via extrinsic binding to its cognate receptor PD-1 expressed on T cells and subsequent downregulation of T cell-mediated anti-tumor responses in general, thereby resulting in a concomitant inhibition of T cell-mediated anti-tumor responses. Recently, intrinsic PD-L1 signaling has been shown to also play a critical role in promoting tumor progression, metastasis, immune evasion, and inducing responsiveness to anti-PD-(L)1 immunotherapy via a PD-1 dependent or independent means [54,55,56,57]. The cell-intrinsic PD-L1 signaling can be modulated by: (**1**) its post-translational modifications such as acetylation or glycosylation, or by forming cis-heterodimers (i.e., the ligand binding to receptors expressed on the same cell), such as PD-L1/PD-1 or PD-L1/CD80 heterodimers (CD80 being a shared receptor for CTLA4 and CD28); (**2**) PD-L1 signaling in cancer cells triggered by engagement of PD-1; and (**3**) PD-L1 binding to co-receptor integrins. From the perspective of biomarker identification, the cell-intrinsic PD-L1 signaling dictates what assays should be used for testing PD-L1 expression. For example, anti-glycosylated PD-L1 specific antibody needs to be used for testing expression of glycosylated PD-L1 on cells, whereas an intracellular staining protocol is needed to assess acetylated PD-L1 located in the cell’s nucleus.

PD-L1 can be cleaved by cell surface proteases such as A Disintegrin. Additionally, Metalloproteinase 10 and 17 (ADAM10 and ADAM17) and the cleaved form of PD-L1 (cPD-L1) can be detected in the serum or plasma of patients with NSCLC, melanoma, and many other cancers [58,59,60]. Several PD-L1 splice variants (sPD-L1) can also be detected in plasma of patients with NSCLC and other cancer types [61,62,63,64]. In addition, the expression levels of PD-L1 in serum or plasma or other components detected in blood include circulating tumor cells (CTCs), exosomes, and various immune cells that have been assessed and proposed to be a predictive biomarker for melanoma, NSCLC, gastrointestinal cancer, breast cancer, and many others [27,65,66,67,68,69,70]. In the following report, we discuss the potential use of PD-L1 expression in different blood components as a predictive biomarker for anti-PD-(L)1 immunotherapy with a focus on addressing the mechanistic and technical challenges that are based on our understanding of the heterogeneity and complexity of the PD-L1 regulation in cancer cells and immune cells. 

## 3. PD-L1 on Circulating Tumor Cells 

Circulating tumor cells (CTCs) are rare cancer cells found in the blood that are derived from solid tumors. The rationale for using PD-L1 positive CTCs (PD-L1^+^CTCs) as a potential biomarker include: (**1**) PD-L1 expressed on CTCs plays an important role in tumor progression and metastasis, as well as resistance of anti-PD-(L)1 immunotherapy [71,72]. (**2**) CTCs derived from primary tumors or metastatic lesions can be detected in blood, providing the opportunity that assessment of the levels of PD-L1 expression in CTCs might recapitulate the PD-L1 expression pattens in tumor tissues [15]. This is important since that it has been shown that the levels of PD-L1 expression are not equal between primary tumors and metastatic tumors in many patients [73]. PD-L1 expression in metastatic tumors, but not in primary tumors, is associated with poor prognosis of melanoma patients [74]. (**3**) The number of CTCs has been shown to be a prognostic biomarker for NSCLC and many other cancers and is also associated with the tumor responses to anti-PD-(L)1 immunotherapy [75,76]. Finally, (**4**) many assays that are currently available to assess the PD-L1^+^CTCs are imperfect and are facing many technical challenges (discussed below). Therefore, assessment of the number of PD-L1^+^CTCs as a biomarker is currently being explored in NSCLC, melanoma, and many other cancers [70,77,78,79]. 

The association between the number of baseline PD-L1^+^CTCs and the efficacy of anti-PD-(L)1 immunotherapy is inconclusive and conflicting. Several studies demonstrate that NSCLC patients with high baseline PD-L1^+^CTCs respond worse to treatment with nivolumab or pembrolizumab [77,79,80,81,82,83]. Similar findings are also reported in melanoma and other types of cancers. However, other studies indicate that baseline numbers of PD-L1^+^CTCs are not associated with patient responsiveness to nivolumab, including NSCLC patients. In addition, some studies demonstrate that the number of PD-L1^+^CTCs is correlated with PD-L1 expression in primary tissues, whereases no correlation was found in other studies [77,80,82,84,85]. Moreover, the detection rates of PD-L1^+^CTCs and its cut-off value in many types of cancers including lung cancer vary dramatically, ranging from less than 30% to 90% across different studies [76,80,84,85,86,87]. The number of PD-L1^+^CTCs from different patients within the same study also vary dramatically, ranging from one cell to hundreds of CTCs in blood samples from prostate cancer patients, or from one cell to 20 in patients with colorectal cancer [84,85,88]. The median percentage of PD-L1 positive cells from different tumors also varies significantly [89].

The inconsistency and heterogeneity of PD-L1^+^CTCs highlight the challenge of reconciling the results of these clinical studies. Among several factors that may contribute to the inconsistency, some of them are listed here. (**1**) Different CTC isolation platforms were used in these studies. It has been shown that the CTC isolation yield differs significantly when different platforms are used [90,91]. Although the CellSearch system (an affinity-based assay for enrichment of EpCAM positive tumor cells) is considered the gold standard CTC detection platform [92], many challenges remain in isolating the rare and highly heterogeneous CTC population [93,94,95]. For example, the CellSearch system cannot be used for isolating EpCAM negative CTCs due to epithelial-to-mesenchymal transition [96]. The procedures and challenges of isolating CTC have been comprehensively reviewed elsewhere [90,95,97,98]. (**2**) The described studies are retrospective analyses with low statistical power due to the limited sample size. (**3**) The results from tumor biopsies demonstrate that PD-L1 is a not a universal biomarker for different lines of therapy. For example, PD-L1 is a predictive biomarker for first-line atezolizumab treatment for NSCLC patients, but the anti-cancer efficacy is not associated with PD-L1 expression for second-line atezolizumab treatment [15,99,100]. However, many studies combined mixed lines of treatments, including first-line to multiple line treatment, making it difficult to interrogate data from these studies. (**4**) The predictive effect of PD-L1 is significantly different among different drugs and cancer types. For example, PD-L1 is a predictive biomarker for first-line and second-line pembrolizumab treatment for NSCLC patients, but not for nivolumab treatment in the same setting [15]. Several studies mentioned above included patients treated with different anti-PD-1 therapeutic antibodies, making it difficult to interrogate data from these studies as well. 

Advances in understanding the biology and regulation of PD-L1 might provide additional opportunities to explore the potential use of PD-L1^+^CTCs as a biomarker. These options include: (**1**) testing PD-L1^+^CTCs from different sources such as obtaining tissues from primary tumors, draining lymph nodes, or tumor metastases. It has already been shown that PD-L1 expression can vary significantly between these different cell sources [97,101,102]. The functions and heterogeneity of PD-L1^+^CTCs suggest that different sources of PD-L1^+^CTCs may be used as a biomarker. (**2**) Testing post-translational modifications of PD-L1, such as acetylation or glycosylation of PD-L1, or its nuclear localization. It has been shown that nuclear expression of PD-L1 may be associated with a poorer prognosis in late stage colorectal or prostate cancer, whereas the nuclear expression is not associated with CTCs. In addition, it has been shown that levels of deglycosylated PD-L1 may have a better predictive value for anti-PD-(L)1 immunotherapy than that of glycosylated PD-L1 when tumor biopsy samples were used [103,104,105,106]. (**3**) CTCs can form CTC clusters. These clusters have shown more metastatic potential than single CTCs in breast cancer and other types of cancers and play a role in cell survival and immune evasion [107,108]. Higher levels of CTC clusters are associated with a worse prognosis in breast cancer patients [109]. In addition, CTCs have also been shown to form clusters with a subset of neutrophils, resulting in the promotion of cancer cell metastasis. Overall, increased levels of CTC–neutrophil clusters are associated with a poorer prognosis in breast cancer patients [110], suggesting that the PD-L1 positive CTCs or CTCs/immune cell clusters might also have a potential to be a biomarker for anti-PD-(L)1 immunotherapy [111,112]. (**4**) Combining the use of PD-L1^+^CTCs with other biomarkers of immune checkpoint receptors. The formation of cis-PD-L1/CD80 heterodimers significantly impacts the efficacy of anti-PD-(L)1 immunotherapy, suggesting that co-expression of CD80 and PD-L1 in CTCs might be a useful biomarker [113]. However, it is unlikely that combining PD-L1^+^CTCs with the approved biomarkers TMB-H or MSI-H would have a predictive value, as multiple studies, based on tumor biopsy results, have demonstrated that PD-L1 is an independent biomarker from TMB-H or MSI-H [114,115,116,117]. Finally, (**5**) PD-L1 gene amplification is associated with patient responsiveness to anti-PD-(L)1 immunotherapy [53,118], and vice versa [119]. Therefore, the level of PD-L1 gene expression in CTCs could be a useful biomarker and is worth pursuing. The rationale and clinical and technical challenges for using PD-L1^+^CTCs are summarized in Table 1 below.

As presented in Table 1, advances in development of a reproducible, accurate, and sensitive method to enrich CTCs regardless of cell sources will facilitate overcoming many of the hurdles that currently exist in the use of PD-L1^+^CTCs, either alone or in combination with other existing biomarkers, to predict treatment responsiveness [91,97,98,120,121,122,123,124,125,126].

## 4. PD-L1 on Circulating Immune Cells

Numerous efforts have been made to explore whether PD-L1 expression on peripheral immune cells can be used as a biomarker for anti-PD-(L)1 immunotherapy. Several lines of evidence support PD-L1 expressed on circulating immune cells as a biomarker for anti-PD-(L)1 immunotherapy. First, nearly all types of immune cells participate in tumor development, progression, metastasis, and resistance to anti-cancer therapies including anti-PD-(L)1 immunotherapy [39,127,128,129,130,131,132,133]. Second, the levels of PD-L1 positive immune cells alone in tumor tissues have prognostic value for many cancer types such as NSCLS, melanoma, renal carcinoma, and many others [134,135]. Moreover, the levels of PD-L1 expression on tumor-infiltrating immune cells or in combination with PD-L1 expression on tumor cells as assessed by IHC are associated with efficacy of atezolizumab or pembrolizumab treatment [15]. Third, the increased microvascular permeability in the tumor tissue promotes the immune cells trafficking to and from the blood stream, suggesting that PD-L1-positive immune cells from tumor tissues can be detected in blood [136,137,138]. Supporting this, it has been shown that T cells expanded in the tumor microenvironment are present in peripheral blood mononuclear cells (PBMCs) [139]. 

The current paradigm regarding the mechanisms underlying anti-PD-(L)1 immunotherapy is that the effective anti-tumor T cell responses are dependent on activation of the dysfunctional tumor-infiltrating CD8^+^ T cells in the tumor microenvironment. Accumulating evidence demonstrates that systemic immunity (defined here as activation of immune responses independent of activation of tumor-infiltrating immune cells) has emerged to be critical for immune-mediated tumor eradication [9,39]. Spitzer et al. demonstrate in a mouse tumor model that activation of T cells in the draining lymph node, bone marrow, and blood contributes to tumor eradication that is independent of the anti-tumor immune response in the tumor microenvironment [140]. Supporting this notion, analyses by single-cell RNA sequencing for T cell receptors in tumor tissues demonstrate that T cells in the tumors are replenished by T cells activated outside of the tumor tissues [139]. Lastly, activated T cells or other immune cells can be found in many organs, including skin, liver, and heart, among others, in patients who develop irAE. Although the origins of these immune cells are not clear, it is possible that activated immune cells from these organs might also be detected in the peripheral blood [141,142,143]. Nonetheless, testing for PD-L1 expression on immune cells in peripheral blood is an attractive approach, as PBMCs might contain both tumor-infiltrating immune cells as well as immune cells activated in the periphery by anti-PD-(L)1 immunotherapy [144,145,146].

### 4.1. PD-L1 Expression on Myeloid Cells

Myeloid cells including neutrophils, monocytes/macrophages, myeloid-derived suppressor cells (MDSCs), and dendritic cells are important modulators of anti-cancer T cell responses [147,148]. The outcomes of these cell-mediated anti-tumor responses are context dependent. For example, macrophages are a type of antigen-presenting cell, but also contribute to a pro-tumor microenvironment. Expression of PD-L1 in these cells in tumor biopsies is associated with the outcomes of patients, as well as patient responses to anti-PD-(L)1 immunotherapy. For biomarker identification, infiltration of myeloid cells has been proposed as a biomarker for anti-PD-(L)1 immunotherapy [149]. We will discuss the opportunities and challenges of assessing expression of PD-L1 in each of these cells as a biomarker for anti-PD-(L)1 immunotherapy. 

#### 4.1.1. Neutrophils

Neutrophils are the most abundant of the white blood cells and are a major part of the innate immune response against infections. The phenotypes and functions of tumor-associated neutrophils (TANs) are highly heterogeneous. TANs display either anti-tumor N1 or pro-tumor N2 phenotype or mixed N1 and N2 phenotypes. These phenotypes are dictated by interaction with tumor cells and other stromal cells in the tumor microenvironment [150,151,152,153,154]. As neutrophil and lymphocyte counts are routinely conducted in clinical settings, the neutrophil-to-lymphocyte ratio (NLR) is one of most studied prognostic biomarkers for cancer progression and anti-cancer therapies including anti-PD-(L)1 immunotherapy [155,156]. The rationale for using NLR is based on multiple studies that demonstrate that the numbers of neutrophils or lymphocytes are associated with poorer, or better outcomes for cancer patients, resulting in amplifying the predictive effects. While many studies demonstrate that cancer patients with higher NLR have a poor prognosis in multiple tumor types or respond poorly to anti-PD-(L)1 immunotherapy, other studies have shown the opposite predictive effects [150,157,158]. To improve its predictive power, combining NLR with other biomarkers has also been explored. A retrospective study by Valero et al. demonstrated that a higher NLR is associated with significantly poorer efficacy of anti-PD-(L)1 immunotherapy in 16 different types of cancers. Combination of NLR with TMB has better predictive effects than NLR alone, and patients that are NLR low/TMB high respond better to anti-PD-(L)1 immunotherapy than patients that are NLR high/TMB high. [159]. Combining NLR with other factors appears to be a better predictive biomarker than NLR alone for NSCLCs patients treated with anti-PD-(L)1 immunotherapy [160,161,162,163,164,165]. In line with the baseline assessment, patients with high NLR after anti-PD-(L)1 immunotherapy respond poorly to anti-PD-(L)1 immunotherapy [166,167]. 

As Mentioned, combining NLR with other factors such as platelet-to lymphocyte ratio appears to improve the predictive abilities [168,169,170]. Conversely, the combination of NLR with PD-L1 tumor proportion score as assessed by IHC or soluble PD-L1 in the blood is inversely associated with the efficacy of anti-PD-(L)1 immunotherapy [171]. PD-L1 positive neutrophils, either alone or in combination with other biomarkers might be worthy of further investigation. 

#### 4.1.2. Monocytes and Macrophages

Macrophages are among the most abundant of immune cells in the tumor microenvironment [172,173]. Tumor-associated macrophages (TAMs) are highly heterogeneous, having shown a mixed anti-tumor M1 phenotype and pro-tumor M2 phenotype. TAMs are differentiated from monocytes and produce numerous growth factors, cytokines, and chemokines to modulate tumor initiation, progression, metastasis, modulation of anti-cancer immune responses, and responses to anti-cancer therapies including PD-(L)1 immunotherapy [133,173]. Several lines of evidence suggest that PD-L1-positive monocytes/macrophages are a potential biomarker for anti-PD-(L)1 immunotherapy. First, multiple studies have shown that most TAMs are M2-like macrophages and that increased numbers of TAMs correlate with a poor prognostic outcome [135,174]. Second, some studies demonstrate that PD-L1 is more highly expressed on immune cells than on tumor cells, and in some cases, PD-L1^+^TAMs are among the most predominant immune cells in the tumor microenvironment [175,176]. Third, while most studies demonstrate that PD-L1^+^ TAMs are immunosuppressive and show pro-tumor phenotypes, other studies suggest that baseline PD-L1^+^ TAMs do not contribute to suppression of anti-tumor T cell responses and in some cases are associated with better prognosis [177,178,179,180]. Similarly, some studies have demonstrated that the numbers of PD-L1^+^ TAMs or PD-L1^+^ monocytes are associated with tumor progression and inversely associated with patient prognosis in multiple cancer types [181,182,183], whereas others studies have demonstrated that baseline numbers of PD-L1^+^ TAMs do not have predictive value for melanoma patients treated with anti-PD-1 immunotherapy [184]. However, in melanoma, the patients who responded better to anti-PD-1 immunotherapy had increased numbers of PD-L1^+^ TAMs [184,185]. Finally, peri-tumor monocytes/macrophages, which could be a potential source of peripheral monocytes, have also been shown to have some prognostic value. Similar to PD-L1^+^ TAMs, it has been shown that peritumor PD-L1^+^ monocytes/macrophages are also immunosuppressive and show a pro-tumor phenotype [181,186]. 

The predictive value of circulating PD-L1^+^ monocytes/macrophages for anti-PD-(L)1 remains less studied and inconclusive. While it has been reported that metastatic melanoma patients with CD14^+^CD16^−^HLA-DR^hi^ monocytes in the peripheral blood prior to treatment respond poorly to anti-PD-1 immune checkpoint therapy [9,187,188], other studies demonstrated that the frequency of PD-L1^+^ CD14^+^ monocytes in the peripheral blood is inversely associated with patient responses to anti-PD-1 immunotherapy [189,190]. Moreover, one study has shown that PD-L1 expression on non-classical (CD14^dim^CD16^+^) and intermediate (CD14^+^CD16^+^) monocytes is significantly increased in patients and is associated with efficacy of melanoma patients treated with anti-PD-1 immunotherapy [191]. 

#### 4.1.3. Myeloid-Derived Suppressor Cells 

Myeloid-derived suppressor cells (MDSCs) are highly immunosuppressive cells derived from immature myeloid progenitor cells [192]. Based on their origin, function, and surface markers, MDSCs are subdivided into two major subsets: polymorphonuclear-MDSCs (PMN-MDSCs) and monocytic MDSCs (M-MDSCs). Cytokines involved in myeloid cell development and differentiation such as GM-CSF, G-CSF, M-CSF, IL-6, and others are major modulators for differentiating myeloid progenitor cells into PMN-MDSCs and M-MDSCs. MDSCs exert their immunosuppressive functions by secreting immunosuppressive factors, including adenosine, IL-10, and TGF-β, as well as increasing activation of regulator T cells. Depletion of MDSCs or targeting MDSCs with pharmaceutical intervention increases the efficacy of anti-PD-(L)1 immunotherapy [192,193,194]. 

PD-L1 is also expressed on PMN-MDSC and M-MDSCs and contributes to MDSC-mediated immunosuppressive effects [195,196,197]. The predictive roles of PD-L1^+^ MDSCs as a biomarker for anti-PD-(L)1 immunotherapy is less studied, although it has been shown that cancer patients with high baseline levels of circulating PMN-MDSC or M-MDSCs respond better to anti-PD-1 immunotherapy treatment [191,198]. It is unlikely that circulating PD-L1^+^MDSCs alone can be a biomarker for anti-PD-(L)1 immunotherapy, as MDSCs produce numerous immunosuppressive factors and cytokines. Thus, PD-L1 expressed on MDSCs may only be partially contributing to MDSC-mediated immunosuppressive effects (i.e., TIM3 ligand Galectin-9 expressed on MDSCs also contributes to MDSC-mediated immunosuppression [199]). 

#### 4.1.4. Dendritic Cells

Dendritic cells (DCs) are the most potent of the antigen-presenting cells and play essential roles in anti-tumor immunity via priming and maintaining effective T-cell-mediated anti-tumor immunity, as well as other innate and adaptive anti-tumor effects [134,200,201,202,203,204]. All subtypes of DCs can be detected in the peripheral blood [190,203]. Based on their morphology, phenotype, and functions, DCs are subdivided into three major subtypes: conventional type 1 dendritic cells (cDC1s), conventional type 2 dendritic cells (cDC2s), and plasmacytoid dendritic cell (pDC) [205,206,207]. Tumor-associated DCs are generally not fully functional and thus cannot truly activate anti-tumor T cell immunity [202,203,204,208,209,210]. Several lines of evidence suggest that PD-L1^+^ DCs might be a potential biomarker for anti-PD-(L)1 immunotherapy. First, the numbers and activation level of tumor-associated DCs have been shown to be a good prognostic indicator in several cancer types [211,212,213,214]. Second, studies from several groups clearly demonstrate that PD-L1 expressed on dendritic cells contributes to inhibition of effective anti-tumor T cell responses. Targeting PD-L1 expressed on DCs using different approaches can, in some cases, stimulate the anti-tumor T cell response to eradicate tumor cells [57,214,215,216,217]. Additionally, PD-L1 intrinsic signaling is involved in DC migration, resulting in a deficiency in T cell priming [218]. Fourth, it has been shown that PD-L1^+^ DCs are associated with a favorable patient outcome and thus may be a good biomarker for stage III colon cancer [219]. Overall, PD-L1 expression on DCs, rather than on tumor cells, appears to be critical for anti-PD-(L)1 treatment-induced anti-tumor immunity [220]. In contrast, another study demonstrated that PD-L1 expressed on cDCs in tumor-draining lymph nodes, but not in the tumor microenvironment, is associated with a poor prognosis in melanoma [221]. 

The predictive roles of circulating PD-L1^+^ cDCs have also not been fully investigated. Liu et al. reported that ovarian cancer and melanoma patients with high expression of PD-L1 on dendritic cells respond better to anti-PD-(L)1 immunotherapy [222]. However, another study reported that patients with high PD-L1 blood DC subtypes respond poorly to PD1 inhibitor therapy [190]. 

### 4.2. PD-L1 Expression on Lymphocytes 

Tumor-infiltrating lymphocytes (TILs) are one of the most studied biomarkers for anti-PD-(L)1 immunotherapy due to the fact that all individual types of lymphocytes (including CD4^+^ T cells, CD8^+^ T cells, B cells, NK cells, and others) have been shown to be critical for effective anti-tumor immunity and that TILs are routinely identified in tumor tissues simply stained with hematoxylin and eosin (H&E) [9,223,224,225,226,227,228,229]. Patients with higher numbers of TILs are generally associated with favorable outcomes in many types of cancers, with a few exceptions, and respond better to anti-PD-(L)1 immunotherapy [229,230]. Of note, the combination of PD-L1 expression in tumor tissues with TILs has also shown a prognostic value for anti-PD-(L)1 immunotherapy [223,224]. 

Nearly all types of TILs were evaluated as a biomarker for anti-PD-(L)1 immunotherapy. Briefly, cancer patients with high numbers of baseline tumor-infiltrating total CD4^+^ T cells, Th1 CD4^+^ T cells, CD8^+^ T cells, the ratio of CD8^+^ T cells/CD4^+^ T cells, T follicular helper-T cells (Tfh), Th1, Th9, and Th17 CD4^+^ T cells respond better to anti-PD-1 immunotherapy, whereas an inverse predictive value was observed in patients with high levels of infiltrating regulatory T cells or Th2 CD4^+^ T cells [226,231]. Further analyzing the association of activation state between TILs with the patient responsiveness to anti-PD-(L)1 immunotherapy indicated that: (1) a higher ratio of tumor-infiltrating Tfh/exhausted CD8^+^ T cells is associated with favorable outcomes for NSCLC patients treated with anti-PD-1 immunotherapy [232]; (2) melanoma patients with a higher percentage of tumor-infiltrating progenitor exhausted CD8^+^ T cells (express intermediate of PD-1, CXCR5), but not terminally exhausted CD8^+^ T cells (express high level of PD-1 and other co-inhibitory receptors such as TIM3), respond better to anti-PD-1 immunotherapy [233]; (3) a combination of signatures of T cell dysfunction and the numbers of infiltrating CD8^+^ T cells has better predictive value than individual PD-L1 levels or TMB for evaluating the efficacy of anti-PD-1 or anti-CTLA-4 immunotherapy [234]; and (4) cancer patients with high levels of tissue-resident memory CD8^+^ T cells (CD8^+^ TRMs) are associated with favorable outcomes and respond better to anti-PD-(L)1 immunotherapy [235].

The predictive values of circulating T cells, in particular circulating PD-L1^+^ T cells, have not been fully investigated, and the results from different studies have yielded different conclusions [236,237,238]. Furthermore, the frequency of circulating PD-L1^+^ T cells (less than 1% of PBMCs for CD3) is significantly lower than for myeloid cells, which poses the technical challenges of developing accurate and robust assays capable of assessing the frequency of circulating PD-L1^+^ T cells. However, it is still worth exploring the predictive value of PD-L1 expression on T cells due to the following: (1) circulating T cells can recapitulate many phenotypes and functional characteristic of TILs, such as matched TCRαβ repertoire, or the gene signatures of effector functions [239]. Moreover, characterization of T cell signatures from patients treated with atezolizumab suggested that activated tumor-infiltrating CD8^+^ T cells may be derived from T cells expanded in the periphery [139]. (2) A number of different types of T cells may be detected in peripheral blood including CD8^+^ TRMs [235], Treg cells [240], and mucosal-associated invariant T cells [241] among others. (3) The frequencies of baseline individual circulating T cell types are associated with responsiveness to anti-PL-(L)1 immunotherapy [241,242]. (4) PD-L1 expressed on T cells has been shown to be a highly immunosuppressive phenotype and can significantly inhibit anti-cancer immunity [243]. Indeed, it has been reported that expression of PD-L1 on circulating T cells is higher in melanoma patients compared to healthy donors, and patients with lower baseline levels of circulating PD-L1^+^ CD4^+^ and CD8^+^ T cells respond better to ipilimumab [244]. 

In addition to demonstrating the role of baseline levels of circulating PD-L1^+^ T cells in ICI-mediated cancer treatments, assessing the levels of circulating PD-L1^+^ T cells after patients are treated with anti-PD-(L)1 immunotherapy has emerged as a new approach to investigate the underlying mechanism and biomarker potential of these molecular markers. The rationale for using on-treatment blood samples is based on the following: (**1**) expression of PD-L1 is upregulated by IFN-γ, and the IFN-γ signature is associated with favorable outcomes for patients treated with anti-PD-(L)1 immunotherapy [245,246,247]; (**2**) treatment with anti-PD-(L)1 immunotherapy significantly increases the numbers of circulating CD4^+^ and CD8 T^+^ cells [248], as well as the frequency of PD-L1^+^ T cells. However, in a dose-escalation phase I trial, it was observed that treatment with avelumab (the only IgG1 mAb; other anti-PD-(L)1 mAbs are all IgG4) did not impact the frequencies of circulating PD-L1^+^CD4^+^ and CD8^+^ T cells in 12 different cancer types. This result needs to be carefully interpreted because of the small sample size (only one patient was evaluated in lung, prostate, ovarian, and other cancers), and the dose of avelumab was not optimized. Given the potential use of the activation state of a patient’s T cells after treatment as a biomarker, we propose that an in vitro PBMC-based assay can be potentially used as a biomarker for anti-PD-(L)1 immunotherapy. For this model, we evaluated T cell functions after nivolumab treatment in the presence of a suboptimal anti-CD3 mAb (non-specific stimulation of TCR signaling). We found that T cell responses to nivolumab vary significantly among different donors. The major rationale for using our model is that T cell responses to nivolumab are similar to those observed in patients treated with anti-PD-(L)1 immunotherapy such as elevated cytokine production, increased proliferative T cells, and increased expression of PD-L1 and many co-inhibitory receptors ([249,250,251,252] and unpublished data), and that activation of T cells by anti-PD-(L)1 immunotherapy might not be tumor antigen-dependent in some cancer patients. Supporting this notion, it was found that PD-L1 is an independent biomarker from TMB in most types of cancers [115]. Renal carcinoma, for example, has a low TMB, but responds well to anti-PD-(L)1 immunotherapy [253].

In addition to T cells, accumulating evidence demonstrates that other types of lymphocytes, including B cells and NK cells as well as tertiary lymphoid structures (TLS, a lymph node-like structure found in the tumor), play critical roles in anti-cancer immunity and are associated with the efficacy of anti-PD-(L)1 immunotherapy. It has been demonstrated that tumor-associated B cells, plasma cells, or TLS are presented in the tumor microenvironment and are associated with improved survival and immunotherapy responsiveness in many types of cancers independent of TILs and PD-L1 expression in tumor tissues [254,255,256,257,258]. Additionally, although the numbers of tumor-infiltrating NK cells (TINKs) is less than T cells and B cells, cancer patients with higher baseline numbers of TINK cells generally have more favorable outcomes and respond better to anti-PD-(L)1 immunotherapy [259,260]. Lastly, different B cell subtypes such as B1B cells, regulatory B cells, also play a role in modulating anti-cancer immunity [261]. Collectively, it is possible that the frequency of circulating NK cells and B cells could also be used for assessing their potential as a predictive biomarker.

The combination of circulating PD-L1^+^ T cells with myeloid cells or tumor cells as a biomarker has also been explored [262]. However, it is challenging to find the right combinations as more available parameters become available. For example, analysis of TILs using mass cytometry has identified 22 different T cell subsets and 17 macrophage subsets in renal carcinoma [263,264]. In another study, a total of 123 immune cell subsets (including circulating PD-L1^+^ immune cells) were identified [265]. Recent advances in artificial intelligence for diagnostic and machine learning techniques will surely provide the opportunity to dissect the complexity and heterogeneity of TILs, as well as dynamic temporal changes of PD-L1 expression on immune cells for biomarker identification [266,267,268,269,270,271]. Expression of PD-L1 on circulating immune cells is unlikely to be useful as a stand-alone biomarker as they are not consistent with those on tumor infiltration immune cells (i.e., TLS only can be found in tumor tissues), and one cannot spatially differentiate tumor-infiltrating PD-L1^+^ immune cells from tumor cells [272,273]. The rationale and clinical and technical challenges in the use of PD-L1-positive immune cells as a biomarker are summarized in Table 2 below.

## 5. Circulating Exosomal PD-L1 

Exosomes are a subset of extracellular membrane-bound vesicles of endosomal origin. Exosomes detected in tumor patients can be derived from all cell types in the primary tumor or metastatic tissues including tumor cells, immune cells, and other stromal cells, such as fibroblasts in the tumor microenvironment, as well as normal cells outside of any tumor [274]. Exosomes serve as a messenger between different cells in the tumor microenvironment by transferring cellular cargoes to the recipient cells [275,276]. Tumor-derived exosomes participate in every stage of cancer development and in modulating anti-cancer therapies via functional proteins, genetic materials such as mRNAs, long non-coding RNAs, and DNA fragments, as well as metabolites [275,276,277,278,279,280,281,282,283]. PD-L1 is highly expressed on tumor-derived exosomes and functions similarly to PD-L1 expressed on tumor cells and immune cells. Exosomal PD-L1 (exoPD-L1) is highly immunosuppressive on T cell-mediated anti-tumor immunity in many types of cancers including melanoma, NSCLC, prostate and breast cancers, and many other types of cancers [284,285,286]. Consistent with the functions of exoPD-L1, the levels of exoPD-L1 are inversely associated with clinical outcome, as reported in several studies [287,288].

As gene expression patterns between tumor tissues and exosomes detected in plasma or serum are closely correlated, and circulating exoPD-L1 (cePD-L1) can be detected at the gene and protein level in many types of cancers, much effort has been devoted to exploring the potential use of circulating exoPD-L1 for monitoring the efficacy of anti-PD-(L)1 immunotherapy [68,289]. Several studies suggest that the baseline cePD-L1 levels are inversely correlated with the efficacy of first- or second-line anti-PD-1 or adjuvant anti-PD-(L)1 immunotherapy in metastatic melanoma patients, NSCLC, and others [285]. Another study indicated that there is no association between the levels of cePD-L1 with the clinical benefits of anti-PD-(L)1 immunotherapy. Combinations of exoPD-L1 with other potential biomarkers were also explored for identifying predictive biomarkers. Zhang et al. reported that patients with high exoPD-L1 and low CD28 expressions respond poorly to anti-PD-1 treatment [290]. In addition, since exosomes also express many immune modulators, such as CD9, CD63, CD81, MHC II, and TGF-β, among others, it is conceivable that the combination of exoPD-L1 with these molecules may identify a better predictive biomarker than exoPD-L1 alone [282,291]. 

Studies on the expression of cePD-L1 and response to anti-PD-(L)1 response remain inconsistent. Several studies have demonstrated that during early stages of anti-PD-1 treatment, there is an increase in the levels of cePD-L1, and increasing magnitudes of cePD-L1 expression are positively associated with better response to the immunotherapy [285,292]. Relevant to this, other studies demonstrated that there is a decrease in the expression levels of PD-L1 mRNA of the plasma-derived exosomes in patients who responded well to anti-PD-1 immunotherapy [66]. Another study suggested that patients with a high increase in cePD-L1 respond poorly to anti-PD-1 immunotherapy, and this is associated with tumor progression [288].

Similar to PD-L1 expressed on tumor cells, there are many technical and mechanistic challenges pertaining to using cePD-L1. The mechanisms and methods for isolating exosomes have been well reviewed elsewhere [293,294], and while significant advances in exosome-based technologies and commercial isolation kits have been developed to increase their sensitivity and accuracy [295,296], there is still a lack of standardized isolation and purification methods for circulating exosomes, especially for characterizing exosomes from different cell origins (tumor cell, immune cells, and fibroblast among others) or from different tumor sites (primary tumor vs. metastatic tumor vs. draining lymph node) [294]. This is important because exoPD-L1 derived from different cell sources functions differently, and this may have different predictive values for anti-PD-(L)1 immunotherapy [280,281]. Additionally, characterization and quantification of cePD-L1 expression by analytical approaches such as flow cytometry-based or ELISA assays among others could be impacted by the method used to isolate the circulating exosomes. Moreover, it has been shown that there is no correlation between exoPD-L1 and PD-L1 assessed by IHC in tumors from NSCLC and melanoma patients. This may be partially due to multiple cell origins of exoPD-L1, and cePD-L1 might be nothing more than an accumulation of exosomal PD-L1 from tumor cells and stromal cells [288,297]. Nonetheless, future work to compare the predictive value between exoPD-L1 and PD-L1 expression in tumors is warranted. The rationale and clinical and technical challenges for using PD-L1^+^CTCs are summarized in Table 3 below.

## 6. Circulating Soluble PD-L1 

The levels of soluble PD-L1 (sPD-L1) in blood have also been explored as a potential biomarker for anti-PD-(L)1 immunotherapy. sPD-L1 is a pool of PD-L1 that may include the cleaved PD-L1 from the surfaces of cells or PD-L1 variants directly released from cells [61,64,298]. Additionally, since most studies used a low-speed centrifugation and did not use exosome isolation protocols, sPD-L1 detected in these studies should also contain cePD-L1. sPD-L1 may also contain PD-L1 mRNA or DNA released from cells in the tumor microenvironment [292,299,300]. Many studies have shown that high levels of sPD-L1 are generally associated with a poor prognosis in many types of cancers, such as lung cancer, melanoma, and renal cell carcinoma, among others [301,302,303,304]. 

However, the predictive value of sPD-L1 for anti-PD-(L)1 immunotherapy remains inconclusive and contradictory. While some studies demonstrated that patients with increased sPD-L1 expression respond better to anti-PD-(L)1 immunotherapy, other studies demonstrated that this increased sPD-L1 is not associated with the efficacy of anti-PD-(L)1 immunotherapy. Several studies have demonstrated that high baseline sPD-L1 levels are associated with a worse response for metastatic NSCLC patients treated with anti-PD-(L)1 immunotherapy [305,306]. A systematic review and meta-analysis of NSCLC patients also indicated that high baseline sPD-L1 levels are associated with a poor response to anti-PD-(L)1 immunotherapy in several cancer types [307]. However, other studies demonstrated that metastatic NSCLC patients with high baseline sPD-L1 levels respond better to first- and later-line nivolumab or Pembrolizumab [308]. Moreover, some studies reported that the baseline levels of sPD-L1 do not correlate with efficacy in NSCLC patients treated with anti-PD-(L1) immunotherapy [309,310,311]. Furthermore, treatment with anti-PD-(L)1 also increases levels of sPD-L1 and PD-L1 mRNA in the blood [292]. 

In summary, most studies have demonstrated that high sPD-L1 levels are associated with a worse response to anti-PD-(L)1 immunotherapy, including NSCLC, in direct contradiction to the notion that high PD-L1 levels in the tumor are a good predictive biomarker for NSCLC. This contradiction may be due to the fact that sPD-L1 contains a mixture of PD-L1 from different cell types within the tumor, different tumor sites, and cells from outside of the tumor as well. 

## 7. Conclusions and Future Perspective 

We have discussed the predictive value of blood-based PD-L1 expression at the cellular and molecular levels based on our evolving understanding of the regulation and function of PD-L1. Although promising, there are a few key issues that need to be explored to further understand the potential use of blood-based PD-L1 as a biomarker, one in which only a causal relationship between blood PD-L1 expression and patient responses to anti-PD-(L)1 immunotherapy is currently observed. 

In addition to the many challenges discussed above, data from many of the currently available studies need to be cautiously interpreted, which is essential for laying a foundation for leveraging the strength of these studies and for the application of artificial intelligence to analyze resulting data. These challenges include the following: (**1**) Several published studies mixed data from patients treated with different lines of therapies (first-, second-, and later-line) and/or different treatment regiments (nivolumab, pembrolizumab, ipilimumab, etc.), and/or from cancer patients in different disease stages, which make current studies less informative. (**2**) While the most significant advantage for blood-based PD-L1 biomarker is its ability to longitudinally monitor the patient’s responses to anti-PD-(L)1 expression, it is important to know that blood drug concentrations have not been analyzed in parallel with testing blood PD-L1 expression in nearly all of these studies. As blood drug levels could significantly affect the accuracy, sensitivity, and specificity of the PD-L1 expression testing results, most of the published studies may not be relevant if drug interference has occurred [312]. (**3**) There are no reliable validated standard assays for testing blood PD-L1 expression. Moreover, most studies lack various negative and/or positive controls to monitor the reliability and robustness of the assay used for testing blood PD-L1 [313]. (**4**) The assays used in most of the studies may not be able to detect different PD-L1 variants [53,280], the post translational forms of PD-L1 [54], or PD-L1 expressed in the nuclear or cytoplasm, which may also impact the interpretation of data. Collectively, a successful development of blood-based PD-L1 as a biomarker needs a well-articulated prospective clinical trial [314,315,316,317,318], a reliable testing platform [319], a sound statistical approach [320,321], and a better understanding of the origin of blood PD-L1 and its potential functions. 

Given the history and challenges of biomarker identification, as well as the heterogeneity of blood PD-L1 expression and regulation, it is unlikely that a blood-based PD-L1 assessment can be used as a stand-alone biomarker for most patients treated with anti-PD-(L)1 immunotherapy. A combination of blood PD-L1 with other biomarker(s) could improve its predictive value for patient stratification. These potential combinations (in addition to combinations discussed above) could be, but are not limited to: (**1**) differential blood-based PD-L1 expression, such as expression of PD-L1 on immune cells, cancer cells, and exosomes; (**2**) other biomarkers representing different mechanisms of action, such as genetic signature [322], immune biomarkers such as serum cytokines, soluble immune regulators (TIM3 and LAG3, etc.), metabolic changes of cancer cells and immune cells [323], epigenetic biomarkers [324], gut microbes, etc. [325,326]; (**3**) tissue-based biomarkers, including PD-L1 and others such as tertiary lymphoid structures; (**4**) in combination with various non-invasive imaging technologies, such as positron emission tomography (PET) imaging, etc. [327,328,329,330,331]. Along with liquid biopsy, the explosion in uses of new omic technologies, such as proteomics, genomics, and multi-omics, provides an unprecedented opportunity for biomarker identification. However, it remains challenging to evaluate the reliability of these assays and to interpret the complex array of data obtained [332,333,334,335]. 

Despite limited success for most proposed biomarkers for anti-PD-(L)1 immunotherapy, we believe that through collaborative efforts by government agencies, industry, and academia [336,337], advances to our deeper understanding of the biology and mechanism of action of anti-PD-(L)1 immunotherapy [127,338,339,340], the application of biomarker-driven clinical trials, efforts to develop robust and sensitive testing technologies, and the application of artificial intelligence for biomarker identification [269,332,341,342] will pave the way for advancing the use of blood-based PD-L1 assessment as part of the armamentarium to deliver the maximal benefits for cancer patients treated with anti-PD-(L)1 immunotherapy.

## Figures and Tables

**Table 1 biomedicines-10-01181-t001:** Clinical and technical challenges in the use of PD-L1^+^CTCs as a biomarker for anti-PD-(L)1 immunotherapy.

Rationale	Clinical Challenges	Technical Challenges	Recent Advances and Trends
Role of CTCs in tumor progression and resistance to anti-PD-(L)1 immunotherapy [71,72] CTCs from tumor tissues can be detected in most cancer patients and have been shown to be associated with prognosis [73] The number of PD-L1^+^CTCs is associated with the clinical efficacy of anti-PD-(L)1 immunotherapy [75,76]	The patient enrollment size for clinical studies is small Not validated by randomized clinical trials using a validated PD-L1^+^CTCs testing platform Clinical trial data are from different lines and different anti-PD-(L)1 antibodies [15,97,98]	Heterogeneity of CTC populations and very low amount of CTCs in circulating blood [93,94,95] Lack of robustness and accuracy in isolating sufficient numbers of CTCs [90,95,97,98]	Testing PD-L1^+^CTCs from different cell/tissue sources [97,101,102] Testing for post-translational modifications of PD-L1 [103,104,105,106] Testing for the PD-L1 positive CTCs/immune cell clusters [111,112] Combination with other biomarkers

**Table 2 biomedicines-10-01181-t002:** Clinical and technical challenges in the use of PD-L1-positive circulating immune cells as a biomarker for anti-PD-(L)1 immunotherapy.

Rationale	Clinical Challenges	Technical Challenges	Recent Advances and Trends
All types of immune cells participate in tumor development, progression, metastasis, and resistance to anti-cancer therapies including anti-PD-(L)1 immunotherapy [39,127,128,129,130,131,132,133] Systemic immunity is critical for immune-mediated tumor eradication [9,39] PBMCs contain both tumor-infiltrating immune cells as well as immune cells activated in the periphery by anti-PD-(L)1 immunotherapy [142,143,144] PD-L1 positive immune cells from tumor tissues can be detected in blood [136,137,138]	The patient enrollment size of clinical studies is small Not validated by randomized clinical trials using validated PD-L1^+^CTCs testing platform The biology of PD-L1 on various subsets of immune cells remains not well understood Conflicting predictions of PD-L1 expression as a biomarker on a variety of immune cells’ response to anti-PD-(L)1 immunotherapy	Lack of standard platform to capture immune cell complexity Heterogeneity of immune cells, as well as dynamic temporal changes of PD-L1 ex-pression on immune cells [266,267,268,269,270,271]	Novel flow cytometry-based assay [263,264] Artificial intelligence for diagnostic and machine learning techniques [266,267,268,269,270,271] Combining PD-L (1) positive immune cells with other biomarkers

**Table 3 biomedicines-10-01181-t003:** Clinical and technical challenges in the use of cePD-L1 as a biomarker for anti-PD-(L)1 immunotherapy.

Rationale	Clinical Challenges	Technical Challenges	Recent Advances and Trends
Derive from all cell types in the primary tumor or metastatic tissues [274] Participate in every stage of cancer development and in modulating anti-cancer therapies [275,276,277,278,279,280,281,282,283] PD-L1 is highly expressed on tumor-derived exosomes and functions similarly to PD-L1 expressed on tumor cells and immune cells [284,285,286] The levels of exoPD-L1 are associated with clinical outcome [287,288] Gene expression patterns between tumor tissues and exosomes detected in plasma or serum are closely correlated [68,289]	The patient enrollment size of clinical studies is small Not validated by randomized clinical trials using validated PD-L1^+^CTCs testing platform No correlation between exoPD-L1 and PD-L1 assessed by IHC in tumors [288,297] Inconsistent data on the correlation of exo/cePD-L1 and response to anti-PD-(L)1 immunotherapies [285,288,292]	Conflicting studies on the expression of exo/cePD-(L)1 [66,283,290] Lack of standardized isolation and purification methods for circulating exosomes [291,292] Characterization of exosomes from different compenents of tumor tissues [292]	Combination of exo/cePD-(L)1 with other biomarkers [280,288,289] Technologies to quantifiy exo/cePD-L1 [278,279,286,295]

## Data Availability

Not applicable.

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
