# Peer review of "Challenges and the Evolving Landscape of Assessing Blood-Based PD-L1 Expression as a Biomarker for Anti-PD-(L)1 Immunotherapy"

_biomedicines, 2022, doi:10.3390/biomedicines10051181_

Round 1

Reviewer 1 Report

The authors did an impressive job with their review that is appreciated.

I have only some minor remarks and suggestions from the perspective of a readership of pathologists: it could be useful to add a paragraph or two on the background points that brought the authors to the decision to focus on the detection of PD-l1 with techniques other than immunohistochemistry. I feel that some background papers on the predictive role of PD-L1 for response to therapy, on the challenges of assessing PD-L1 with immunohistochemistry, and on the potential of molecular pathology on specimens other than histological slides are worth quoting and discuss:

-Burtness et al 2022, doi 10.1200/JCO.21.02198

-Haddad et al 2022, doi 10.1136/jitc-2021-003026

-Emancipator et al 2021, doi 10.1038/s41379-020-00710-9

-Torlakovic et al 2020, doi 10.1038/s41379-019-0327-4

-Girolami et al 2021, doi 10.1111/jop.13220

-Pepe et al 2020, doi 10.1002/cncy.22400

-Satturwar et al 2022, doi 10.1002/dc.24955

Author Response

The authors did an impressive job with their review that is appreciated. I have only some minor remarks and suggestions from the perspective of a readership of pathologists: it could be useful to add a paragraph or two on the background points that brought the authors to the decision to focus on the detection of PD-l1 with techniques other than immunohistochemistry. I feel that some background papers on the predictive role of PD-L1 for response to therapy, on the challenges of assessing PD-L1 with immunohistochemistry, and on the potential of molecular pathology on specimens other than histological slides are worth quoting and discuss:

Responses: We appreciate the positive and constructive comments from the reviewer. As recommended, we discuss in greater detail the challenges of pathological assessment of PD-L1 expression in tumor tissues as a predictive biomarker.

Reviewer 2 Report

The manuscript entitled:" Challenges and the evolving landscape of assessing blood-based PD-L1 expression as a biomarker for anti-PD-(L) immunotherapy" focused on a systemic revision of literature data about the novel approaches available to detect PD-L1 expression in torrent blood is well written and requires some minor revisions to be suitable for publication

  • In the manuscript, i would suggest to also describe the most suitable approaches usefull to isolate CTC. In my opinion, this aspect may improve the readibility of the manuscript at the sight of tehnical aspects related to the isolation of this cell population.
  • In the text, the authors report the clinical strategiers available in the patients positive for PD-L1 expression. As regards, please, could the authors enrich this section by elucidating the clinical setting where ICIs may be approached in the clinical setting?
  • In the text, the authors dedicate a paragraph to PD-L1 evaluated on exosomes. Please, could the authors also underline the compelx approach followed to purify exosomes? Could this aspect imapct on the pD-L1 characterization?
  • Please, could the authors include a table that summarizes technical and clinical aspects described in the manuscript?

Author Response

The manuscript entitled:" Challenges and the evolving landscape of assessing blood-based PD-L1 expression as a biomarker for anti-PD-(L) immunotherapy" focused on a systemic revision of literature data about the novel approaches available to detect PD-L1 expression in torrent blood is well written and requires some minor revisions to be suitable for publication

We appreciate the constructive comments from the reviewers. We are herein providing a point-by-point response to all their comments:

In the manuscript, i would suggest to also describe the most suitable approaches useful to isolate CTC. In my opinion, this aspect may improve the readability of the manuscript at the sight of technical aspects related to the isolation of this cell population.

Responses:

We have provided, as suggested, additional information on the isolation methods of CTC as well as some of their pros and cons.  The associated references have also been added to the revised manuscript.

In the text, the authors report the clinical strategies available in the patients positive for PD-L1 expression. As regards, please, could the authors enrich this section by elucidating the clinical setting where ICIs may be approached in the clinical setting?

Responses:

We have added to the paper a discussion revolving around how tumor biopsy-based PD-L1 expression is currently used for patient stratification in the clinic for many cancers. Although we think that blood-based PD-L1 has potential as a biomarker in the clinic setting, any such use would need to be validated in clinical trials. Therefore, we remain cautious on the global applications of blood-based PD-L1 as a clinical biomarker.

In the text, the authors dedicate a paragraph to PD-L1 evaluated on exosomes. Please, could the authors also underline the complex approach followed to purify exosomes? Could this aspect impact on the pD-L1 characterization?

Responses:

We agree that characterization of exosomal PD-L1 can be directly impacted by different exosomal PD-L1 analytical approaches following the purification of exosomes. Due to the complexity of this issue, we briefly discuss these issues in the revised manuscript and provide additional, technically-focused references for perusal.

Please, could the authors include a table that summarizes technical and clinical aspects described in the manuscript?

Responses:

As suggested by this reviewer, we have added three separate tables summarizing the technical and clinical aspects reviewed in this MS.